# Hybrid Techniques for the Diagnosis of Acute Lymphoblastic Leukemia Based on Fusion of CNN Features

**DOI:** 10.3390/diagnostics13061026

**Published:** 2023-03-08

**Authors:** Ibrahim Abdulrab Ahmed, Ebrahim Mohammed Senan, Hamzeh Salameh Ahmad Shatnawi, Ziad Mohammad Alkhraisha, Mamoun Mohammad Ali Al-Azzam

**Affiliations:** 1Computer Department, Applied College, Najran University, Najran 66462, Saudi Arabia; 2Department of Artificial Intelligence, Faculty of Computer Science and Information Technology, Alrazi University, Sana’a, Yemen

**Keywords:** CNN, RF, XGBoost, ALL, PCA, hybrid method

## Abstract

Acute lymphoblastic leukemia (ALL) is one of the deadliest forms of leukemia due to the bone marrow producing many white blood cells (WBC). ALL is one of the most common types of cancer in children and adults. Doctors determine the treatment of leukemia according to its stages and its spread in the body. Doctors rely on analyzing blood samples under a microscope. Pathologists face challenges, such as the similarity between infected and normal WBC in the early stages. Manual diagnosis is prone to errors, differences of opinion, and the lack of experienced pathologists compared to the number of patients. Thus, computer-assisted systems play an essential role in assisting pathologists in the early detection of ALL. In this study, systems with high efficiency and high accuracy were developed to analyze the images of C-NMC 2019 and ALL-IDB2 datasets. In all proposed systems, blood micrographs were improved and then fed to the active contour method to extract WBC-only regions for further analysis by three CNN models (DenseNet121, ResNet50, and MobileNet). The first strategy for analyzing ALL images of the two datasets is the hybrid technique of CNN-RF and CNN-XGBoost. DenseNet121, ResNet50, and MobileNet models extract deep feature maps. CNN models produce high features with redundant and non-significant features. So, CNN deep feature maps were fed to the Principal Component Analysis (PCA) method to select highly representative features and sent to RF and XGBoost classifiers for classification due to the high similarity between infected and normal WBC in early stages. Thus, the strategy for analyzing ALL images using serially fused features of CNN models. The deep feature maps of DenseNet121-ResNet50, ResNet50-MobileNet, DenseNet121-MobileNet, and DenseNet121-ResNet50-MobileNet were merged and then classified by RF classifiers and XGBoost. The RF classifier with fused features for DenseNet121-ResNet50-MobileNet reached an AUC of 99.1%, accuracy of 98.8%, sensitivity of 98.45%, precision of 98.7%, and specificity of 98.85% for the C-NMC 2019 dataset. With the ALL-IDB2 dataset, hybrid systems achieved 100% results for AUC, accuracy, sensitivity, precision, and specificity.

## 1. Introduction

Blood is the dynamic engine and one of the basic elements of the human body, which consists of three main components with different weights: red blood cells (45%), plasma (55%), and white blood cells (WBC) (less than 1%) [1]. These components differ according to their color, shape, texture, composition, size, and any increase or decrease in the percentage of each element in the blood that causes a specific disease [2]. Leukemia is a deadly blood cancer that produces a malignant WBC that attacks normal blood cells and causes death [3]. It is one of the most common cancers among children and adults. The bone marrow produces the basic components of blood [4]. Therefore, malignant WBC affects the bone marrow due to its ability to make the basic components of blood normally and causes a weakening of the immune system. In addition, malignant WBC travels through the bloodstream and causes damage to the liver, spleen, kidneys, brain, etc., and causes other forms of fatal diseases [5]. The affected WBC type determines the type of leukemia: lymphoid (Acute myeloidleukemia) if the cells are monocytes, or myelogenous (acute lymphoblastic leukemia) if the cells are lymphocytes [6]. Acute lymphoblastic leukemia (ALL) is classified into three forms according to its shape and size: L1, L2, and L3 as shown in Figure 1. Type L1 cells are symmetrical in shape, of a regular small round size, and surrounded by little cytoplasm. Type L2 cells are asymmetric in form, irregular, larger than L1, and have different cytoplasm. L3-type cells have a normal shape and size, an oval or circular nucleus, its size is larger than L1, and a quantity of cytoplasm surrounds it with vacuoles [7]. When lymphoid or myeloid cells grow abnormally and uncontrollably, they cause leukemia. These cancer cells reduce the possibility of the growth of normal blood cells in the bone marrow [8]. The few normal blood cells are discharged into the bloodstream which cannot supply the body’s organs with sufficient oxygen, so that the immune system will weaken and the blood clotting will be weak. ALL represents 25% of childhood cancers; 74% of leukemia cases in people under 20 have ALL. The five-year survival rate for children under 14 is 91%, while people between the ages of 15 and 20 have a five-year survival rate of 75%. In all cases diagnosed early and treated, the ALL disease cannot return, meaning that children diagnosed with ALL, after five years, became healthy and recovered completely. However, the recovery rate for adults five years after their injury is not high, ranging between 20–35% [9]. ALL spreads quickly if left undiagnosed and can lead to death within months. ALL is usually diagnosed by a complete blood count test, in which a doctor checks for certain clinical signs of leukemia [10]. Sometimes the doctor is unsatisfied with the symptoms of a complete blood count smear to decide on leukemia. Therefore, the doctor resorts to suctioning a sample of bone marrow and examining it under a microscope to confirm the presence of leukemia. All manual methods for diagnosing leukemia depend on the experience of specialists and doctors. The procedures for manual leukemia diagnosis are difficult, complex, prone to human error, costly, and time-consuming. To overcome the limitations of manual diagnosis, it would be beneficial to automate the examination processes of blood and bone marrow samples. Deep and machine-learning networks have proven their ability to address the limitations of manual diagnosis for early detection of ALL. Machine-learning algorithms are highly capable of classifying handcrafted attributes. Pre-processing is one of the most important steps of AI to improve images and remove artefacts that degrade system performance. Additionally, the blood slice images contain many blood components and non-target cells, so the segmentation method works to crop white blood cells only, called regions of interest (ROI), which are sent to feature extraction methods to extract features of white blood cells only. In recent years, CNN networks have inputted the medical field, which has high capabilities to identify many diseases, including distinguishing between normal and blasted blood cells. CNN automatically learns the hierarchy of spatial features from the input images. CNN has millions of parameters, biases, weights, and connections that can be learned and adapted while training the data. The weights and parameters are adjusted by minimizing the difference between the actual and expected data through the backpropagation of the network. The main motivation of this work is to develop effective microscopic blood slide analysis models for the diagnosis of ALL. The medical dataset lacks huge images, making it difficult to train the CNN network from scratch so that pre-trained CNN models can extract features from the C-NMC 2019 and ALL_IDB2 datasets. To improve the performance of the proposed systems, PCA was used to select only important features, combine features of CNN, and feed them to RF and XGBoost algorithms.

The main major contributions of this study are as follows:Enhancement of microscopic blood images by two consecutive filters;Applying the active contour algorithm to only extract WBC regions and feed them to CNN models;Analysis and diagnosis of ALL images using hybrid techniques of CNN-RF and CNN-XGBoost;Selection of highly representative features and removal of redundant ones using PCA;Fusion of deep feature maps of CNN models to obtain hybrid deep feature map vectors for DenseNet121-ResNet50, ResNet50-MobileNet, DenseNet121-MobileNet, and DenseNet121-ResNet50-MobileNet models and their classification by RF and XGBoost classifiers.

The rest of the paper is organized as follows: Section 2 analyzes previous studies’ techniques for diagnosing ALL and summarizes their findings. Section 3 explains proposed strategies for analyzing microscopic blood images to diagnose ALL. Section 4 summarizes the results of the proposed methods. Section 5 discusses the performances of all systems and compares them. Section 6 concludes the study.

## 2. Related Work

This section presents a collection of previous studies in which researchers focused on diagnosing glass slide images for ALL detection.

Ghada et al. [11] used a CNN based on a Bayesian approach to analyze microscopic smear images to diagnose ALL disease. The Bayesian approach frequently searches for optimal parameters to minimize objective error. Muhammad et al. [12] used a VGG16 model based on Efficient Channel Attention (ECA) to extract deep features for better classification. The ECA works to overcome the similarities between natural and explosive images. The model achieved an accuracy of 91.1% for diagnosing the C-NMC dataset. Niranjana et al. [13] converted images to HIS color space, segmented WBC cells, then trained the ALLNET model and tested its performance, which achieved an accuracy of 95.54% and a sensitivity of 95.91%. Pradeep et al. [14] used three CNN models to extract image features of an ALL dataset and classify them by RF and SVM. Sorayya et al. [15] modified the weights and parameters of the ResNet50 and VGG16 models to train the ALL dataset. They also proposed six machine-learning algorithms and a convolutional network with ten convolutional layers and a classification layer. The convolutional network achieved an accuracy of 82.1%, while the VGG16 network achieved an accuracy of 84.62%. The best accuracy of the machine-learning algorithm was 81.72% by RF. Gundepudi et al. [16] used the AlexNet hybrid model with four machine-learning algorithms to analyze microscopic blood images to classify the ALL-IDB2 dataset. AlexNet extracts the features and feeds them into machine-learning algorithms for classification. Raheel et al. [17] used a hybrid of two CNNs for detecting ALL. The background was removed, noise was reduced, and cells of interest were segmented. The leukemia dataset features were extracted by two CNNs and combined and classified by SVM. Mohamed et al. [18] used the obstetric adversarial network to classify blood samples for the ALL-IDB dataset and evaluate the results through a hematologist. Rana et al. [19] used heat mapping and a PCA assessment of prediction of whole blood cell count by ANN which helped to increase the accuracy of diagnosis of morphometric parameters of leukemia samples. A heat map fed with cell population data produced a cluster to separate bone marrow from lymphoblastic leukemia. The network achieved an accuracy of 89.4%. Sanam et al. [20] used a generative adversarial algorithm to validate data augmentation, feeding the leukemia dataset to the CNN model based on Tversky loss function with controlled convolution and density layers. Tulasi et al. [21] used the GBHSV–Leuk method for segmenting and classifying ALL cell diseases. The technique consists of two stages: the Gaussian Blurring filter to improve the images and the Hue Saturation Value technique to separate the cell from the rest of the image. The GBHSV–Leuk method achieved an accuracy of 95.41%. Yunfei et al. [22] used the ternary stream-driven, WT-DFN-augmented data classification network to identify lymphoblasts. An attention map is generated for each image to highlight infected cells. Distinctive features are obtained by erasing and cropping attention. Luis et al. [23] used the LeukNet model based on the VGG16 model with fewer dense layers. Network parameters were adjusted to evaluate the leukemia dataset, which achieved an accuracy of 82.46%. Mohamed et al. [24] used a DNN hybrid network trained on CLL MRD from 202 F-DNN patients and 138 L-DNN patients. DNN proved its ability to detect CLL MRD with an overall accuracy of 97.1%. Ibrahim et al. [25] performed an extraction of handcrafted features and their classifications by FFNN and SVM. They also used a hybrid technique between CNN and SVM to classify the two datasets, ALL_IDB1 and ALL_IDB2.

Jan et al. [26] used a flow cytometry method which detects and analyzes specific genetic drift through machine-learning algorithms such as decision trees and gradient boosting. The deviation appears in t(12;21)/ETV6-RUNX1 blast cells with high CD10 and CD34 and low CD81 expression. Nada et al. [27] used a machine-learning algorithm for acute leukemia classification based on a feature-selection algorithm by the gray-wolf optimization method. Adaptive thresholding was applied to improve images and then classify them by SVM, KNN, and NB. The SVM achieved an accuracy of 96%, a sensitivity of 89.5%, and an accuracy of 94.5%.

Ahmad et al. [28] proposed four machine-learning algorithms for image analysis of the C-NMC dataset for leukemia prediction. The images were optimized and features were extracted using three DNN models. ANOVA analyzed data, then the features were selected by random forest. The SVM algorithm achieved the best accuracy of 90% compared with other algorithms.

The diversity of methodologies and researchers focused on reaching satisfactory results for the early detection of ALL disease is noted from previous studies. Thus, this study focused on extracting highly representative features by applying active contour technology to obtain the WBC region and sending them to CNN models for analysis. This research also focused on extracting deep feature maps for CNN models and merging feature maps of multiple models. Finally, the RF and XGBoost classifiers classify the fused CNN features.

## 3. Materials and Methods

### 3.1. Description of Datasets

The systems proposed in this study were evaluated using two datasets. The C-NMC 2019 dataset is for developing strategies and testing their performance. The second dataset is ALL_IDB2 which evaluates the extent to which systems are generalized to classify any future dataset.

#### 3.1.1. Description of C-NMC 2019 Dataset

Systems were trained and evaluated by the C-NMC 2019 dataset. This dataset was released from the Cancer Imaging Archive (TCIA) for the ALL Disease Diagnostics Competition. TCIA is funded by the Cancer Program of the National Cancer Institute in the United States of America, which aims to develop computer-assisted technologies. The ALL Disease Diagnostics Competition aims to design automated systems to distinguish blasted blood cells (leukemia) from normal cells. The dataset consisted of 10,661 blood clotting micrographs divided into 7272 images from 47 leukemic subjects and 3389 images from 26 healthy subjects. All dataset images are in RGB color space with a 24-bit resolution of 450 × 450 pixels [29].

#### 3.1.2. Description of ALL_IDB2 Dataset

The performance of the developed systems was evaluated on the ALL-IDB2 dataset (ALL-International Data Base), which is available online and contains the deadliest images of ALL. Disease experts classified all images as acute or normal leukemia. The dataset images were acquired with a Canon PowerShot G5 microscope and saved in RGB color space with a 24-bit resolution of 1944 × 2592 pixels. The dataset consisted of 260 microscopic blood images of coagulation divided into 130 images of patients with leukemia and 130 images of healthy subjects [30].

### 3.2. Improving Microscopic Blood Images

The first step in the proposed systems is to improve the images to remove the artifacts in the image, which are considered as features if they are not removed. Optimization techniques help to obtain good results for image diagnosis [31]. Microscopic blood images contain artifacts caused by varying microscopes, light reflections, and solutions that mix with blood samples. In this study, the mean RGB colors were adjusted and color constancy was measured. Subsequently, microscopic blood images were fed into the average filter to clean them of noise and artifacts [32]. The averaging filter factor was set to 6 × 6, which means that in each rotation, one pixel called the target is optimized based on calculating the average of neighboring pixels (summing the value of 35 pixels and dividing it by 35). The filter was repeated to process each image pixel, as shown in Equation (1). Then, the images enhanced by the average filter were fed to the Laplacian filter to show the edges of the WBC cells and improve the low contrast between the WBC cells and the cytoplasm, as shown in Equation (2) [33].
(1)$$Y\left(x\right)=\frac{1}{N}\overset{N-1}{\underset{j=0}{\sum }}z\left(x-j\right)$$
where $Y\left(x\right)$ means the input, $z\left(x-i\right)$ means the previous input, and *N* means number of pixels.
(2)$$\nabla {\;}^{2}\;f\left(m,n\right)=\frac{\partial {\;}^{2}\;f}{\partial^{\;}m_{2}}+\frac{\partial {\;}^{2}\;f}{\partial^{\;}n_{2}}$$
where $\nabla {\;}^{2}\;f$ means the differential equation as second-order and *m*, *n* means pixels’ location.

Figure 2 shows samples of images of the C-NMC 2019 dataset after improvement, while Figure 3 shows samples of images of the ALL_IDB2 dataset after improvement.

### 3.3. Active Contour Technique

The segmentation stage is a tedious and important process in the medical imaging processing stages. It is the most difficult stage and it must be accurate and efficient because the following medical imaging processing steps depend on segmentation accuracy. Segmenting is challenging due to low contrast, reflection, color variation, irregular borders, and asymmetry. Blood micrographs contain two parts: WBC cells and cytoplasm, and other blood components, so fractionation is the separation of WBC cells for further processing. The segmentation process has information about the region of interest, not the full picture. If segmentation is done accurately, it will extract highly representative features. Several segmentation algorithms are based on the intensity of similar pixels, thresholding, and detecting WBC cell boundaries and embedded regions [34].

This study applied a powerful segmentation algorithm called the active contour model to separate WBC cells from the rest of the blood components. The active contour is used in many fields, including medical image segmentation, to segment desired regions (pixels) for analysis. The algorithm separates WBC cells from the background for further processing. Active perimeter is defined as a model that separates the ROI in pixels from the background to obtain high classification results. The algorithm has a superior ability to extract the boundaries of WBC cells because it provides a smooth contour. A contour is a border to select WBC cells; it is a set of points subject to the interpolation process, which is polylinear to define the object’s boundaries. Active contour is a process for obtaining deformable structures with constraints and forces in segmentation blood microscopy images, which describe WBC cell boundaries and other features to form contour and curve parameters [35]. The curve models are determined by applying intrinsic and extrinsic forces through Equation (3). These forces are related to the curves of the images of microscopic cells. External energy is the combination of energy that controls the circumference of WBC cells within the image. Internal energy controls deformable changes [36].
(3)$$G_{cv}C=\underset{outside}{\int }{\left(I\left(x\right)-m_{\;1}\right)}^{2}\;dx+\underset{inside}{\int }{\left(I\left(x\right)-m_{\;2}\right)}^{2}\;dx+\beta \;length\;\left(C\right)$$
where *I* refers to gray intensity, *outside* and *inside* refer to the regions outside and inside the contour *C*, *m*_1_ and *m*_2_ refer to mean intensity, outside and inside respectively, *β* refers to parameter, and *length*(*C*) refers to the length of contour *C*. *G_cv_* has the sensitivity for initializing and few parameters to set. *G* makes the contour move to the boundary of the WBC cell.

The active contour technique extracts WBC cell regions for all images of the C-NMC 2019 and ALL-IDB2 datasets. Then, it saves them in a new folder to feed them into CNN models and traditional feature extraction methods, as shown in Figure 4.

### 3.4. Strategy of Machine-Learning with Fusion Features CNN

The pre-trained models on the ImageNet dataset face some challenges, such as not reaching satisfactory accuracy because the ImageNet dataset does not contain medical images. Additionally, CNN models need a long time for implementation and need high-level computers. Therefore, this study presented a hybrid strategy based on feature extraction by CNN and replacing classification layers with machine-learning algorithms [37]. It is worth noting that CNN models received images of the C-NMC 2019 and ALL-IDB2 datasets after cropping WBC cells (ROI). Convolutional and pooling layers help extract deep feature maps and convert them into feature vectors. RF and XGBoost classifiers separately receive feature vectors for DenseNet121, ResNet50, and MobileNet.

#### 3.4.1. Extraction of High-Level Features by CNN

This study developed hybrid systems between deep and machine-learning based on segmentation techniques of regions of interest to separate WBC cells from other blood components. The microscopic blood images were optimized and fed into active contour technology to extract ROI (WBC cells). Then, ROI of the C-NMC 2019 and ALL-IDB2 datasets was fed into three CNN models for feature extraction. Features were extracted using DenseNet121, ResNet50, and MobileNet. Then, features of more than one model were combined and classified using RF and XGBoost classifiers.

Convolutional neural networks are more accurate, deeper, and more efficient in training due to the connection of neurons with previous layers and in the same layer [38]. CNN models connect each layer with the following layers in a feed-forward manner. For each layer, the feature maps of the previous layers were used as input and the feature maps of the current layers were produced as output. The CNN model has many advantages: it makes features more widespread and diversified through convolutional layers, reduces the number of parameters by pooling layers, and reduces overfitting problems through dropout layers [39].

The dense connection pattern of DenseNet121 requires fewer parameters than convolutional networks, where redundant features are not re-learned. The DenseNet121 architecture can distinguish between information added to the form and information saved. DenseNet121 layers are very narrow; each layer consists of many feature maps and adds feature maps to the collective knowledge while keeping the feature maps unchanged. In addition to parameter efficiency, the DenseNet121 network has the important advantage of gradients and information flow through the network layers, which facilitates network training. DenseNet121 leads to deep tacit supervision and in-depth training, reducing overfitting due to minority classes [40]. Each layer in DenseNet121 can directly access the gradients from the original input-loss function. Feature maps that learn from different layers increase variance in subsequent layers and improve model training performance [41]. Each layer adds new special feature maps. The network growth rate is the extent to which adding information contributes to that layer for reaching the global state [42].

ResNet50 is a residual module belonging to the ResNetxx family, consisting of 16 blocks distributed among 177 layers. ResNet50 is one of the most important and accurate CNN models in many fields of computer vision. ResNet50 takes the input images in the input layer, resizes the images to 224 × 224 pixels, and transfers them to the next layers. The model contains 49 convolutional layers with filters of different sizes to extract the features by wrapping the filter *f* (*t*) around the image *x* (*t*) as in Equation (4) [43]. The pooling layers receive huge amounts of neurons, which require complex operations that take a long time, and these layers work to reduce the high dimensions [44]. The model contains average pooling layers that calculate the average of the selected neurons and replace the selected neurons with their average value as in Equation (5). The model also contains max-pooling layers, which work to select a group of neurons and replace them with a single neuron that has the max value as in Equation (6). There are auxiliary layers, such as ReLU, for further processing of features, where the positive is passed and the negative is converted to zero. Dropout layers prevent overfitting issues by stopping 50% of neurons each time and passing 50% [45].
(4)$$W\left(t\right)=\left(x\;*\;f\right)\;\left(t\right)=\int x\left(a\right)f\left(t-a\right)\;\;da$$*W* (*t*), *x* (*t*), and *f* (*t*) and refer to the output, the inputted image, and the filter, respectively.
(5)$$z\left(i;\;j\right)=max_{m,n=1\ldots .k}\;f\left[\left(i-1\right)p+m;\;\left(\;j-1\right)p+n\right]$$
(6)$$z\left(i;\;j\right)=\frac{1}{k^{2}}\underset{m,n=1\ldots .k}{\sum }f\left[\left(i-1\right)p+m;\;\left(\;j-1\right)p+n\right]$$
where *f* means size of filter, *m*, *n* is the matrix location, *p* means Filter wrap, and *k* means the vectors.

MobileNet is a lightweight network architecture based on depth-wise convolutions. Depth-wise is a form of separable convolution, where the standard layers are divided into two layers depth-wise, and the 1 × 1 convolution is called point convolution. Depth-wise separable convolutions apply convolutions to a single input channel [46]. Point-wise convolution applies a 1 × 1 convolution to merge the output into depth-wise convolution layers. Standard convolution layers work with both filters and merge the inputs into a new set of outputs. The depth-wise separable convolution is divided into two layers: a separate layer for filtering purposes and a separate layer for combining purposes [47]. This factor has the effect of reducing the computation and size of the model. MobileNet architecture contains 28 layers (depth-wise and point-wise layers) followed by ReLU and batchnorm [48].

Finally, the deep feature maps are extracted from the last layers of DenseNet121, ResNet50, and MobileNet as follows: (16, 32, 512), (3, 3, 512), and (7, 7, 1024), respectively. Therefore, highly represented features are converted into vector features by global average pooling, which produces vector feature vectors of sizes 1024, 2048, and 1024 for DenseNet121, ResNet50, and MobileNet, respectively. The features for the C-NMC 2019 dataset are represented in a feature matrix of size 10,661 × 1024, 10,661 × 2048, and 10,661 × 1024 for DenseNet121, ResNet50, and MobileNet models, respectively. The features for the ALL-IDB2 dataset are represented in a feature matrix of size 260 × 1024, 260 × 2048, and 260 × 1024 for DenseNet121, ResNet50, and MobileNet models, respectively.

#### 3.4.2. Random Forest Classifier

Random Forest is a highly efficient machine-learning classifier for solving classification problems. The RF is based on ensemble learning by combining the results of several classifiers and making a decision based on the majority vote. The classifier’s name suggests the many decision tree classifiers from subsets of the original dataset and takes the average prediction accuracy for all trees rather than relying on a single tree [49]. The higher the number of the decision tree, the more efficient and reliable the accuracy. The following steps show how the algorithm works. First, randomly select L data points from the training dataset. Second, create decision trees with the specified number of data points. Third, separately train the decision trees and compute the average majority vote. Fourth, for the test data points, the predictions of all decision trees are searched and the algorithm will classify the test data point into the class that wins the majority of votes. Ensemble learning models work in two methods, Bagging and Boosting [50].

The RF algorithm works with a Bagging method that creates a subset of the training dataset, replacement, and final decision based on a majority vote. The algorithm starts with random data from the training set known as Bootstrap data. This process is known as Bootstrapping. Models are individually trained, which leads to multiple outcomes called aggregation. Finally, the individual results are combined and the final decision is based on a majority vote.

#### 3.4.3. XGBoost Classifier

XGBoost is a powerful machine-learning classifier that is based on gradient-boosted decision trees. The algorithm creates a decision tree that is sequentially based on assigning weights to independent variables and inserting them into the decision trees to predict results. The algorithm integrates weak learners into strong learners through sequential models to produce a strong model. This process is called Boosting. When training the network, the weights of the variables predicted by the method (decision tree) are incorrectly increased [51]. The variables are sequentially fed to the next decision tree. Boosting is an ensemble learning method for creating strong classifiers from serially weak classifiers. The first model is built to predict the training dataset, then, building the next predictive model to solve the errors in the previous model, and the process continues until the predicted training dataset is accurate [52].

This hybrid strategy has two techniques. The first technique is shown in Figure 5, which goes through the following sequence: first, optimizing the images of the C-NMC 2019 and ALL-IDB2 datasets. Second, extracting WBC cells and isolating them from other major blood components by the active contour method and keeping them in a folder in a new dataset called ALL-IDB2-ROI. Third, the C-NMC 2019 and ALL-IDB2-ROI datasets are fed into DenseNet121, ResNet50, and MobileNet models to extract high-level feature maps and then represent them into feature vectors with sizes 10,661 × 1024, 10,661 × 2048, and 10,661 × 1024 for DenseNet121, ResNet50, and MobileNet models, respectively, for C-NMC 2019 dataset. Whereas with the ALL-IDB2 dataset, features were saved in feature vectors with sizes 260 × 1024, 260 × 20,148, and 1024 × 470 for DenseNet121, ResNet50, and MobileNet models, respectively.

Fourth, dimensionality reduction of high feature vectors by PCA and retention of accurate representative features in feature vectors of sizes of 10,661 × 440, 10,661 × 630, and 10,661 × 470 for DenseNet121, ResNet50, and MobileNet models, respectively, for C-NMC 2019 dataset. Whereas with the ALL-IDB2 dataset, features were saved in feature vectors with sizes 260 × 440, 260 × 630, and 260 × 470 for DenseNet121, ResNet50, and MobileNet models, respectively. Fifth, the RF and XGBoost classifiers separately receive feature vectors of the DenseNet121, ResNet50, and MobileNet models.

As shown in Figure 6, the second technique goes through the following sequence: the first four steps are the same as the steps of the first technique. Fifth, serially combining feature vectors into new hybrid vectors as follows: combine vectors of DenseNet121-ResNet50, ResNet50-MobileNet, DenseNet121-MobileNet, and DenseNet121-ResNet50-MobileNet into new feature vectors with sizes of 10,661 × 1070, 10,661 × 1100, 10,661 × 910, and 10,661 × 1540 for C-NMC 2019 dataset. Whereas with the ALL-IDB2 dataset, features were saved in feature vectors of sizes 260 × 1070, 260 × 1100, 260 × 910, and 260 × 1540. Sixth, the RF and XGBoost classifiers receive the CNN mixed feature vectors and classify them with high precision.

## 4. Results of System Evaluation

### 4.1. Splitting the C-NMC 2019 and ALL_IDB2 Datasets

This study developed several hybrid systems of deep and machine-learning based on hybrid features from two datasets, C-NMC 2019 and ALL-IDB2, for acute leukemia. The C-NMC 2019 and ALL-IDB2 datasets contain 10,661 and 260 micrographs, respectively. It is worth noting that the C-NMC 2019 dataset contains microscopic images of WBC cells only, while the ALL-IDB2 dataset contains microscopic images of WBC blood cells along with other essential blood components. During the implementation of the systems, the two datasets were divided into 80% for training the systems and validation (80:20) and feeding it with sufficient information for its readiness to generalize it to any other future dataset, and 20% for testing, as shown in Table 1.

### 4.2. Systems Evaluation Metrics and Hyperparameters

The confusion matrix is the core criterion for evaluating systems’ performance on a medical dataset. In this study, the systems received the images of the two datasets and produced the confusion matrix as an evaluation tool. The confusion matrix is in the form of a quadrilateral matrix. The output classes are represented by rows, while the columns represent the target classes. Confusion matrix cells contain all test pictures that the systems correctly classify as TP and TN and pictures that the systems incorrectly classify as FP and FN [53]. Therefore, the performance of the methods in this study was measured through Equations (7)–(11).
(7)$$\mathrm{AUC}\;=\frac{\mathrm{TP}\;\mathrm{Rate}}{\mathrm{FP}\;\mathrm{Rate}}$$
(8)$$\mathrm{Accuracy}=\frac{\mathrm{TN}+\mathrm{TP}}{\mathrm{TN}+\mathrm{TP}+\mathrm{FN}+\mathrm{FP}}\;*\;100\%$$
(9)$$\mathrm{Sensitivity}=\frac{\mathrm{TP}}{\mathrm{TP}+\mathrm{FN}}\;*\;100\%$$
(10)$$\mathrm{Precision}=\frac{\mathrm{TP}}{\mathrm{TP}+\mathrm{FP}}\;*\;100\%$$
(11)$$\mathrm{Specificity}=\frac{\mathrm{TN}}{\mathrm{TN}+\mathrm{FP}}\;*\;100$$

Table 2 shows the hyperparameter options tuning during model training for the two acute leukemia datasets.

### 4.3. Augmentation Data Technique for Balancing Classes

Achieving satisfactory results with deep learning models requires a large number of images during the training phase to perform well during the testing phase, generalize the systems to another dataset, and avoid overfitting. Unbalanced dataset classes lead to tendencies of accuracy to majority classes, which poses a challenge to artificial intelligence systems. The data augmentation technique addressed these challenges facing deep learning models. This technology solves both challenges in parallel [54]. To avoid the problem of overfitting and the extent to which systems generalize to other datasets, the data was artificially increased so that new images were artificially created from the same dataset through many operations such as rotating at different angles, shifting up and down, flipping, and others. To solve the problem of class imbalance, the same technique was used, where the images of the majority class were increased by a lesser amount than the minority classes; thus, this technique solved the two challenges in parallel as shown in Table 3.

### 4.4. Results of CNN Model

This section presents the performances of the DenseNet121, ResNet50, and MobileNet models for analyzing microscopic blood images for diagnosing two acute leukemia datasets. The DenseNet121, ResNet50, and MobileNet models received acute leukemia images and extracted feature maps through convolutional layers. Fully connected layers received high-level feature maps and put them into single-level feature vectors. The SoftMax activation function labeled each image into its appropriate class and classified all images into ALL or normal classes.

The DenseNet121, ResNet50, and MobileNet models yielded good results for classifying the C-NMC 2019 and ALL-IDB2 acute leukemia datasets, as shown in Table 4 and Figure 7.

First, for the C-NMC 2019 dataset, DenseNet121 achieved an AUC of 94.65%, accuracy of 90.3%, sensitivity of 88.8%, precision of 89%, and specificity of 88.35%. ResNet50 achieved an AUC of 95.15%, accuracy of 90.8%, sensitivity of 90.55%, precision of 90.5%, and specificity of 90.53%. In contrast, the MobileNet model achieved an AUC of 93.05%, accuracy of 89.4%, sensitivity of 87.95%, precision of 87.7%, and specificity of 88.3%.

Second, for the ALL-IDB2 dataset, DenseNet121 achieved an AUC of 93.8%, accuracy of 90.4%, sensitivity of 90.15%, precision of 90.45%, and specificity of 90.2%. ResNet50 achieved an AUC of 92.35%, accuracy of 90.3%, sensitivity of 92.25%, precision of 92.55%, and specificity of 91.8%. In contrast, MobileNet achieved an AUC of 93.05%, accuracy of 90.4%, sensitivity of 90.25%, precision of 90.45%, and specificity of 89.95%.

### 4.5. Results of CNN Model Based on Segmentation Algorithm

This section presents the performances of DenseNet121, ResNet50, and MobileNet models based on WBC region segmentation to analyze blood micrographs diagnosing two acute leukemia datasets. DenseNet121, ResNet50, and MobileNet models received improved images and fed them into the active contour algorithm to isolate the WBC region from the rest of the blood components. The DenseNet121, ResNet50, and MobileNet models received images of regions of interest (WBC cells) and extracted feature maps through convolutional layers. Fully connected layers received high-level feature maps and put them into single-level feature vectors. The SoftMax activation function labeled each image in its appropriate category and classified all images into ALL or normal classes.

Based on the segmentation algorithm, the DenseNet121, ResNet50, and MobileNet models yielded good results for classifying the C-NMC 2019 and ALL-IDB2 acute leukemia datasets, as shown in Table 5 and Figure 8.

First, for the C-NMC 2019 dataset, DenseNet121 with features of WBC region achieved an AUC of 95.95%, accuracy of 94%, sensitivity of 93.3%, precision of 93.15%, and specificity of 92.75%. ResNet50 with features of WBC region achieved an AUC of 94.9%, accuracy of 93.5%, sensitivity of 93.1%, precision of 92.7%, and specificity of 92.85%. In contrast, MobileNet with features of WBC region achieved an AUC of 96.5%, accuracy of 94.2%, sensitivity of 93.25%, precision of 93.4%, and specificity of 93.05%.

Second, for the ALL-IDB2 dataset, DenseNet121 with features of WBC region achieved an AUC of 96.95%, accuracy of 84.2%, sensitivity of 93.65%, precision of 94.85%, and specificity of 93.9%. ResNet50 with features of WBC region achieved an AUC of 97.1%, accuracy of 96.2%, sensitivity of 95.8%, precision of 96.45%, and specificity of 95.75%. In contrast, MobileNet with features of WBC region achieved an AUC of 96.65%, accuracy of 96.2%, sensitivity of 96.1%, precision of 96.15%, and specificity of 96.35%.

### 4.6. Results of Strategy of Machine-Learning with Features CNN

This section presents the performance of hybrid systems between deep and machine-learning for analyzing microscopic blood images for diagnosing two acute leukemia datasets. The mechanism of action of this technique is to segment the WBC cell region and isolate it from the rest of the basic blood components. The DenseNet121, ResNet50, and MobileNet models received the region of interest, extracted the deep feature maps, reduced the dimensions, and saved the high representation features by PCA. RF and XGBoost classifiers received highly representative features and classified them with high accuracy.

The RF and XGBoost classifiers with features DenseNet121, ResNet50, and MobileNet yielded good results for classifying acute leukemia’s C-NMC 2019 and ALL-IDB2 datasets.

First, for the C-NMC 2019 dataset, as shown in Table 6 and Figure 9, the RF classifier with DenseNet121 features achieved an AUC of 98.15%, accuracy of 95%, sensitivity of 94.8%, precision of 94.15%, and specificity of 94.3%. In contrast, RF achieved with ResNet50 features achieved an AUC of 97.25%, accuracy of 95.5%, sensitivity of 95.35%, precision of 94.8%, and specificity of 94.7%. RF with MobileNet features achieved an AUC of 96.75%, accuracy of 96.2%, sensitivity of 95.75%, precision of 95.55%, and specificity of 95.65%.

The XGBoost classifier with DenseNet121 features achieved an AUC of 95.5%, accuracy of 94.7%, sensitivity of 96.1%, precision of 93.7%, and specificity of 93.8%. In contrast, XGBoost achieved with ResNet50 features achieved an AUC of 97.65%, accuracy of 95.4%, sensitivity of 94.15%, precision of 95.25%, and specificity of 94.3%. XGBoost with MobileNet features achieved an AUC of 96%, accuracy of 95.3%, sensitivity of 94.35%, precision of 94.45%, and specificity of 94.75%.

Second, for the ALL-IDB2 dataset, RF and XGBoost classifiers with CNN-fused features achieved promising results. The RF classifier with features of DenseNet121, ResNet50, and MobileNet achieved accuracy, AUC, sensitivity, precision, and specificity of 100%.

In contrast, the XGBoost classifier with features of DenseNet121, ResNet50, and MobileNet achieved accuracy, AUC, sensitivity, precision, and specificity of 100%.

When evaluating the systems for the dataset C-NMC 2019: The RF classifier with CNN features produces a confusion matrix, as shown in Figure 10. The hybrid systems DenseNet121-RF, ResNet50-RF, and MobileNet-RF achieved good results for each class. First, DenseNet121-RF attained an accuracy of 96.1% and 92.6 for ALL and normal classes. Second, ResNet50-RF attained an accuracy of 96.6% and 93.4% for ALL and normal classes. Third, MobileNet-RF attained an accuracy of 97.1% and 94.2% for ALL and normal classes.

Figure 11 shows the confusion matrix generated by the hybrid systems DenseNet121-XGBoost, ResNet50-XGBoost, and MobileNet-XGBoost. First, DenseNet121-XGBoost attained an accuracy of 95.8% and 92.2% for ALL and normal classes. Second, ResNet50-XGBoost attained an accuracy of 97.7% and 90.4% for ALL and normal classes. Third, MobileNet-XGBoost attained an accuracy of 96.3% and 93.2% for ALL and normal classes.

When evaluating the systems for the ALL-IDB2 dataset: the RF classifier with CNN features produces a confusion matrix, as shown in Figure 12. The hybrid systems DenseNet121-RF, ResNet50-RF, and MobileNet-RF achieved good results at each class level. First, DenseNet121-RF attained an accuracy of 100% and 100% for ALL and normal classes. Second, ResNet50-RF attained an accuracy of 100% and 100% for ALL and normal classes. Third, MobileNet-RF attained an accuracy of 100% and 100% for ALL and normal classes.

Figure 13 shows the confusion matrix generated by the hybrid systems DenseNet121-XGBoost, ResNet50-XGBoost, and MobileNet-XGBoost. First, DenseNet121-XGBoost attained an accuracy of 100% and 100% for ALL and normal classes. Second, ResNet50-XGBoost attained an accuracy of 100% and 100% for ALL and normal classes. Third, MobileNet-XGBoost attained an accuracy of 100% and 100% for ALL and normal classes.

### 4.7. Results of Strategy of Machine-Learning with Fusion Features CNN

This section presents the performances of hybrid systems based on hybrid CNN features for analyzing blood micrographs for diagnosing two acute leukemia datasets. The mechanism of action of this technique is to improve microscopic blood images, then segment the WBC cell region and isolate it from the rest of the main blood components. DenseNet121, ResNet50, and MobileNet models received a region of interest (WBC cells), extracted feature maps, and then selected the important features and deleted duplicates by PCA. The deep feature maps between CNN models were serially merged as follows: DenseNet121-ResNet50, ResNet50-MobileNet, DenseNet121-MobileNet, and DenseNet121-ResNet50-MobileNet. The hybrid CNN features were fed to RF and XGBoost classifiers to classify them accurately.

The RF and XGBoost classifiers with CNN fusion features yielded superior results for classifying the C-NMC 2019 and ALL-IDB2 acute leukemia datasets.

First, for the C-NMC 2019 dataset, RF and XGBoost classifiers with CNN-fused features achieved promising results as shown in Table 7 and Figure 14. The RF classifier with fusion features of DenseNet121-ResNet50 achieved an AUC of 98.1%, accuracy of 97.7%, sensitivity of 97%, precision of 97.75%, and specificity of 96.8%. The RF classifier with fusion features of ResNet50-MobileNet achieved an AUC of 97.4%, accuracy of 97.3%, sensitivity of 97.15%, precision of 96.95%, and specificity of 97.35%. In contrast, the RF classifier with fusion features of DenseNet121-MobileNet achieved an AUC of 98.55%, accuracy of 98.5%, sensitivity of 98.05%, precision of 98.4%, and specificity of 98.25%. The RF classifier with fusion features of DenseNet121-ResNet50-MobileNet achieved an AUC of 99.1%, accuracy of 98.8%, sensitivity of 98.45%, precision of 98.7%, and specificity of 98.85%.

In contrast, the XGBoost classifier with fusion features of DenseNet121-ResNet50 achieved an AUC of 97.05%, accuracy of 96.3%, sensitivity of 96.1%, precision of 95.45%, and specificity of 95.8%.

The XGBoost classifier with fusion features of ResNet50-MobileNet achieved an AUC of 97.85%, accuracy of 97.6%, sensitivity of 97.8%, precision of 97.15%, and specificity of 97.35%. In contrast, the XGBoost classifier with fusion features of DenseNet121-MobileNet achieved an AUC of 98.85%, accuracy of 98.1%, sensitivity of 97.8%, precision of 97.85%, and specificity of 97.95%. The XGBoost classifier with fusion features of DenseNet121-ResNet50-MobileNet achieved an AUC of 98.55%, accuracy of 98.2%, sensitivity of 98%, precision of 97.85%, and specificity of 98.15%.

Second, for the ALL-IDB2 dataset, RF and XGBoost classifiers with CNN-fused features achieved promising results. The RF classifier with fusion features of DenseNet121-ResNet50, ResNet50-MobileNet, DenseNet121-MobileNet, and DenseNet121-ResNet50-MobileNet achieved accuracy, AUC, sensitivity, precision, and specificity of 100%.

In contrast, the XGBoost classifier with fusion features of DenseNet121-ResNet50, ResNet50-MobileNet, DenseNet121-MobileNet, and DenseNet121-ResNet50-MobileNet achieved accuracy, AUC, sensitivity, precision, and specificity of 100%.

When evaluating dataset systems C-NMC 2019: The RF classifier with fusion CNN features produces a confusion matrix, as shown in Figure 15. First, DenseNet121-ResNet50-RF attained an accuracy of 98.9% and 95.3% for ALL and normal classes. Second, ResNet50-MobileNet-RF attained an accuracy of 98.1% and 95.7% for ALL and normal classes. Third, DenseNet121-MobileNet-RF attained an accuracy of 99.2% and 97.1% for ALL and normal classes. Fourth, DenseNet121-ResNet50-MobileNet-RF attained an accuracy of 99.3% and 97.6% for ALL and normal classes.

The XGBoost classifier with fusion CNN features produces a confusion matrix, as shown in Figure 16. First, DenseNet121-ResNet50-XGBoost achieved an accuracy of 96.7% and 95.4% for ALL and normal classes. Second, ResNet50-MobileNet-XGBoost achieved an accuracy of 98% and 96.8 for ALL and normal classes. Third, DenseNet121-MobileNet-XGBoost achieved an accuracy of 98.7% and 96.8% for ALL and normal classes. Fourth, DenseNet121-ResNet50-MobileNet-XGBoost achieved an accuracy of 98.6% and 97.5% for ALL and normal classes.

When evaluating dataset systems ALL-IDB2: The RF classifier with fusion CNN features produces a confusion matrix, as shown in Figure 17. First, DenseNet121-ResNet50-RF attained an accuracy of 100% and 100% for ALL and normal classes. Second, ResNet50-MobileNet-RF attained an accuracy of 100% and 100% for ALL and normal classes. Third, DenseNet121-MobileNet-RF attained an accuracy of 100% and 100% for ALL and normal classes. Fourth, DenseNet121-ResNet50-MobileNet-RF attained an accuracy of 100% and 100% for ALL and normal classes.

The XGBoost classifier with fusion CNN features produces a confusion matrix, as shown in Figure 18. First, DenseNet121-ResNet50-XGBoost achieved an accuracy of 100% and 100% for ALL and normal classes. Second, ResNet50-MobileNet-XGBoost achieved an accuracy of 100% and 100% for ALL and normal classes. Third, DenseNet121-MobileNet-XGBoost achieved an accuracy of 100% and 100% for ALL and normal classes. Fourth, DenseNet121-ResNet50-MobileNet-XGBoost achieved an accuracy of 100% and 100% for ALL and normal classes.

## 5. Discussion of the Results of the Performances of the Systems

Acute leukemia is fatal for children and young adults if not diagnosed early. The characteristics of the blasted white blood cells are similar to normal, especially in the first stage, which causes a challenge for hematologists. Therefore, computer-aided automated systems help in distinguishing infected WBC from normal. This work focused on developing systems consisting of two parts, deep and machine-learning, based on fused features. Microscopic blood images were improved for all systems and WBCs were isolated from the rest of the main blood components using the active contour method.

The first strategy was to evaluate the pre-trained DenseNet121, ResNet50, and MobileNet models of the C-NMC 2019 and ALL-IDB2 acute leukemia datasets. With the C-NMC 2019 dataset, DenseNet121, ResNet50, and MobileNet achieved an accuracy of 90.3%, 91.8%, and 89.4%, respectively. Whereas with the ALL-IDB2 dataset, the DenseNet121, ResNet50, and MobileNet models achieved 90.4%, 92.3%, and 90.3% accuracy, respectively.

The second strategy for analysis of blood micrographs for diagnosing of NMC 2019 and ALL-IDB2 datasets using DenseNet121, ResNet50, and MobileNet models was based on the active contour method. With the C-NMC 2019 dataset, DenseNet121, ResNet50, and MobileNet achieved an accuracy of 94%, 94.2%, and 93.5%, respectively. Whereas with the ALL-IDB2 dataset, the DenseNet121, ResNet50, and MobileNet models achieved 94.2%, 96.2%, and 96.2% accuracy, respectively.

The third strategy was diagnosing images of two NMC 2019 and ALL-IDB2 datasets using hybrid techniques between CNN models (DenseNet121, ResNet50, and MobileNet) and RF and XGBoost classifiers based on the active contour method. With the C-NMC 2019 dataset, the DenseNet121-RF, ResNet50-RF, and MobileNet-RF systems achieved an accuracy of 100%, 100%, and 100%, respectively. In contrast, the DenseNet121-XGBoost, ResNet50-XGBoost, and MobileNet-XGBoost systems achieved an accuracy of 95%, 95.5%, and 96.2%, respectively. Whereas with the ALL-IDB2 dataset, the DenseNet121-RF, ResNet50-RF, and MobileNet-RF systems achieved an accuracy of 100%, 100%, and 100%, respectively. In contrast, DenseNet121-XGBoost, ResNet50-XGBoost, and MobileNet-XGBoost achieved an accuracy of 100%, 100%, and 100%, respectively.

The fourth strategy was to diagnose images of two NMC 2019 and ALL-IDB2 datasets by hybrid techniques between CNN models (DenseNet121, ResNet50 and MobileNet) and RF and XGBoost classifiers based on fused CNN features. With the C-NMC 2019 dataset, the DenseNet121-ResNet50-RF, ResNet50-MobileNet-RF, DenseNet121-MobileNet-RF, and DenseNet121-ResNet50-MobileNet-RF systems achieved accuracy of 97.7%, 97.3%, 98.5%, and 98.8%, respectively. In contrast, the DenseNet121-ResNet50-XGBoost, ResNet50-MobileNet-XGBoost, DenseNet121-MobileNet-XGBoost, and DenseNet121-ResNet50-MobileNet-XGBoost systems achieved accuracy of 96.3%, 97.6%, 98.1%, and 98.2%, respectively. With the ALL-IDB2 dataset, the DenseNet121-ResNet50-RF, ResNet50-MobileNet-RF, DenseNet121-MobileNet-RF, and DenseNet121-ResNet50-MobileNet-RF systems achieved accuracy of 100%, 100%, 100%, and 100%, respectively. In contrast, the DenseNet121-ResNet50-XGBoost, ResNet50-MobileNet-XGBoost, DenseNet121-MobileNet-XGBoost, and DenseNet121-ResNet50-MobileNet-XGBoost systems achieved accuracy of 100%, 100%, 100%, and 100%, respectively.

Table 8 and Figure 19 summarize the results of the proposed systems for analyzing microscopic blood images of the two C-NMC 2019 and ALL-IDB2 acute leukemia datasets. The table shows each system’s overall accuracy and each class’s accuracy for each system. It is noted that the pre-trained CNN models did not produce promising results for the detection of acute leukemia. While the results improved when the images were optimized, the region of interest (WBC) was segmented and then fed to the CNN models for classification. The pre-trained CNN models on the ImageNet dataset do not achieve superior results because the ImageNet dataset on which the models are trained lacks medical images. Therefore, hybrid systems CNN-RF and CNN-XGBoost were applied, where it was noticed that the results improved and the systems achieved superior results for diagnosing the two leukemia datasets. It is noted that the accuracy is better improved when the CNN features are serially combined and classified with RF and XGBoost classifiers.

For the C-NMC 2019 dataset, RF with fused features of DenseNet121-ResNet50-MobileNet reached the best accuracy for class ALL and normal with 99.3% and 97.6%, respectively. For the ALL-IDB2 dataset, the CNN-RF and CNN-XGBoost systems achieved an accuracy of 100% for both ALL and normal classes. The RF and XGBoost classifiers with fused features of CNN achieved an accuracy of 100% for ALL and normal classes.

## 6. Conclusions

Acute leukemia is one of the deadliest forms of leukemia affecting children and adults. It is treated through early detection before spreading to other body parts. Early detection of acute leukemia is essential for obtaining appropriate treatment. This research contributed to developing effective strategies for early diagnosis of acute leukemia by analyzing the images of C-NMC 2019 and ALL-IDB2 leukemia datasets. The blood micrographs were optimized and fed into the active contour method to extract only the WBC region for analysis by CNN models. This study focused on WBC area analysis, feature extraction by CNN models (DenseNet121, ResNet50, and MobileNet), and classification using RF and XGBoost classifiers. Each CNN model produces high and redundant features, therefore, PCA was applied to select the most important features and delete the redundant ones. Due to the similarity of the characteristics of infected and normal WBC in the early stages, and to obtain more efficient features, the CNN features were serially combined as follows: DenseNet121-ResNet50, ResNet50-MobileNet, DenseNet121-MobileNet, and DenseNet121-ResNet50-MobileNet and classified using RF and XGBoost classifiers. The systems have achieved promising results and have proven effective in assisting hematologists in the early detection of acute leukemia. RF classifier based on the fused features of DenseNet121-ResNet50-MobileNet achieved an AUC of 99.1%, accuracy of 98.8%, sensitivity of 98.45%, precision of 98.7%, and specificity of 98.85%.

The most important limitation faced in this study is the low number of images in the dataset, which causes overfitting, which was overcome by the data augmentation technique.

In future works, systems for diagnosing acute leukemia by hybrid techniques will be developed based on combining features of CNN models with color, shape, and texture features extracted by traditional feature extraction methods.

## Figures and Tables

**Figure 1 diagnostics-13-01026-f001:**
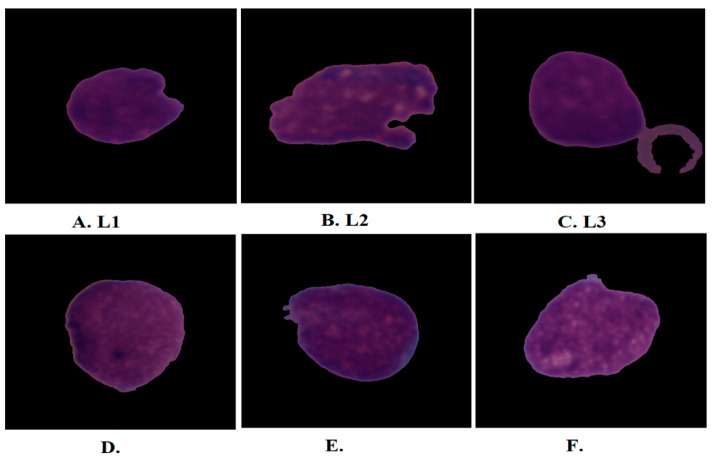
Samples from the C-NMC 2019 dataset. (**A**–**C**) acute lymphoblastic leukemia while (**D**–**F**) healthy images.

**Figure 2 diagnostics-13-01026-f002:**
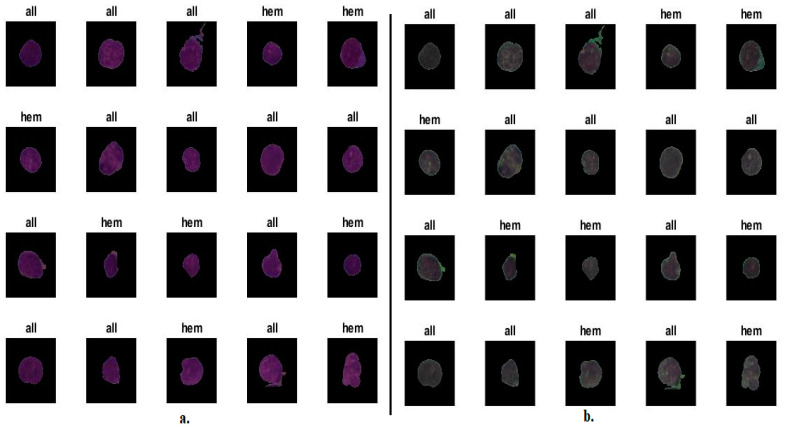
Random selection of images from the C-NMC 2019 dataset (**a**). Before improvement. (**b**) After improvement.

**Figure 3 diagnostics-13-01026-f003:**
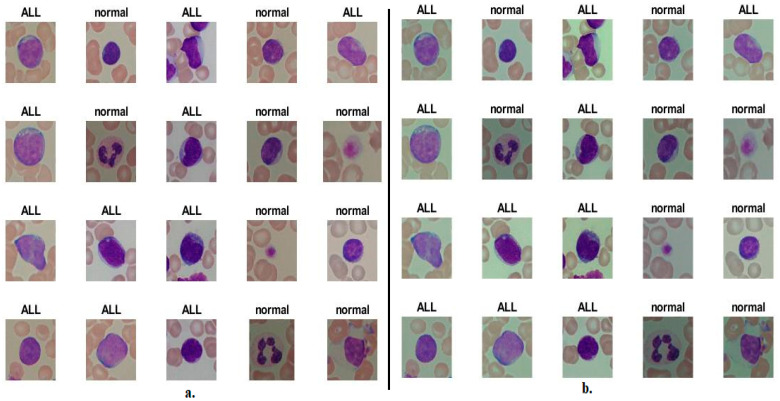
A random selection of images from the ALL_IDB2 dataset (**a**). Before improvement. (**b**) After improvement.

**Figure 4 diagnostics-13-01026-f004:**
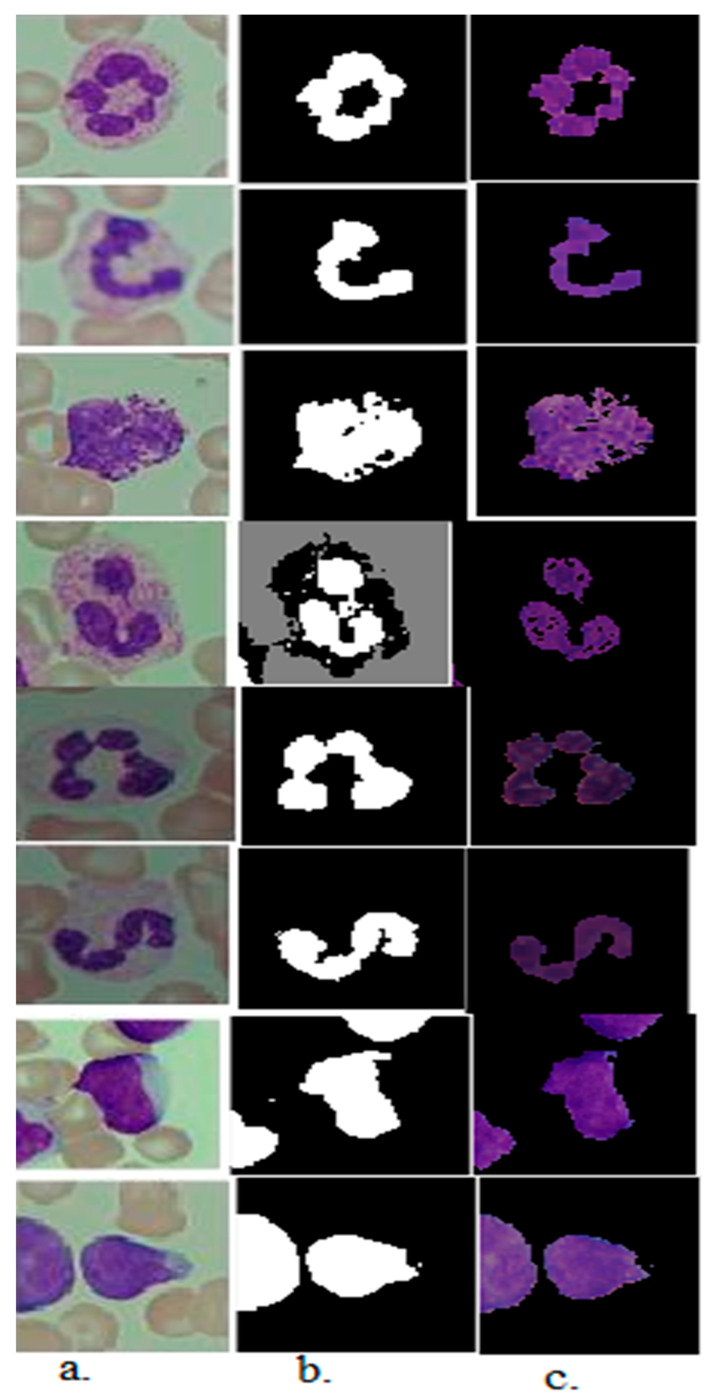
A set of samples from the acute leukemia dataset. After enhancement. (**a**) Original images (**b**) Segmentation of WBC cells. (**c**) Region of interest (WBC cells).

**Figure 5 diagnostics-13-01026-f005:**
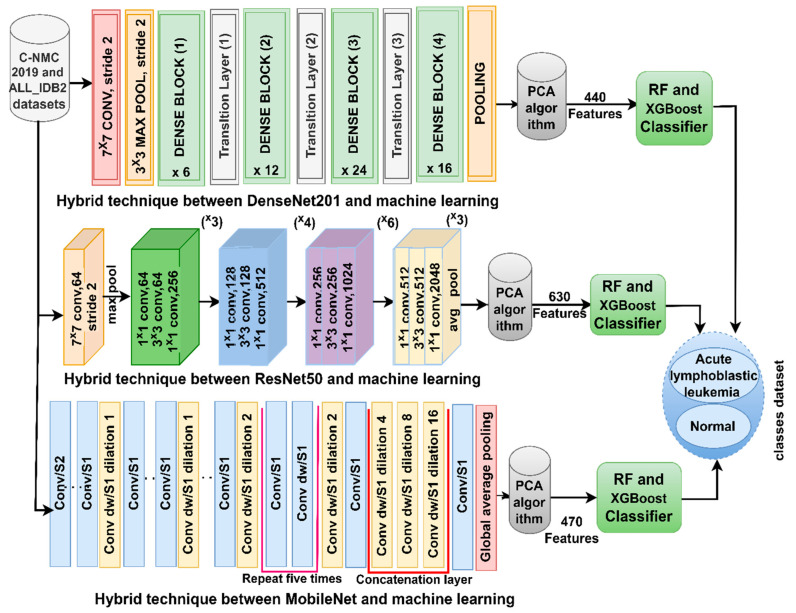
Framework of a hybrid technique between CNN and deep learning for the diagnosis of ALL.

**Figure 6 diagnostics-13-01026-f006:**
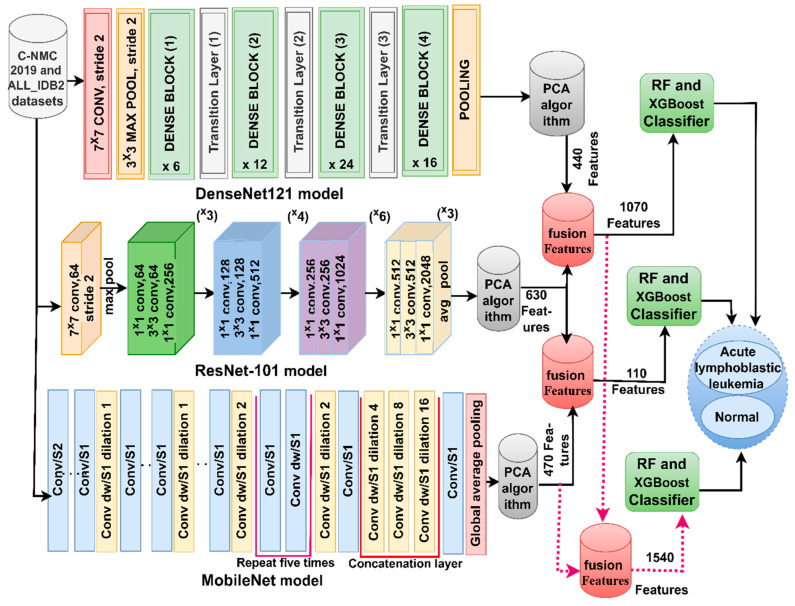
Hybrid technique framework based on hybrid CNN features for the diagnosis of ALL.

**Figure 7 diagnostics-13-01026-f007:**
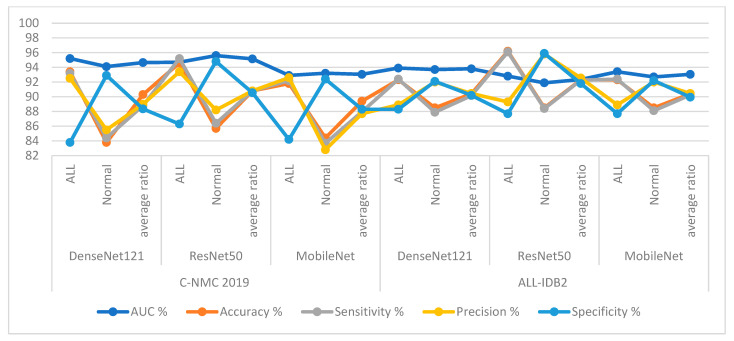
Display of results of CNN models of the C-NMC 2019 and ALL-IDB2 datasets for acute leukemia diagnosis.

**Figure 8 diagnostics-13-01026-f008:**
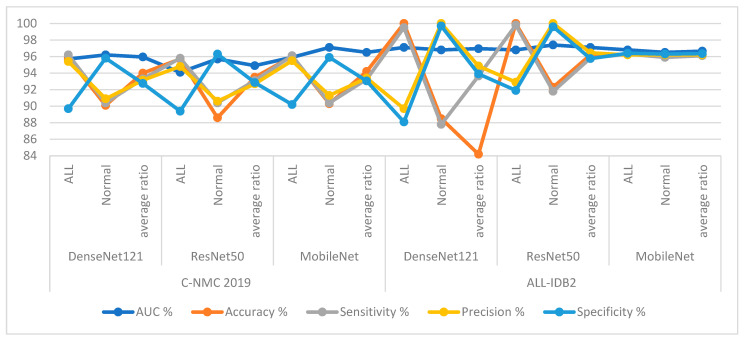
Display of results of CNN models based on WBC segmentation of the C-NMC 2019 and ALL-IDB2 datasets for acute leukemia diagnosis.

**Figure 9 diagnostics-13-01026-f009:**
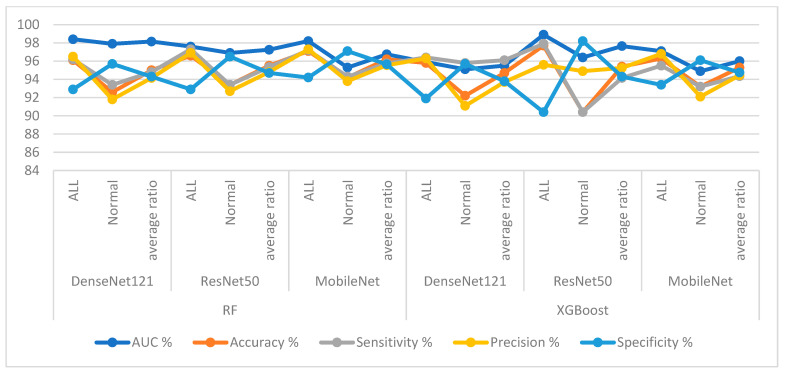
Display of results of the RF and XGBoost classifiers based on the features of the CNN models of the C-NMC 2019 dataset for the diagnosis of acute leukemia.

**Figure 10 diagnostics-13-01026-f010:**
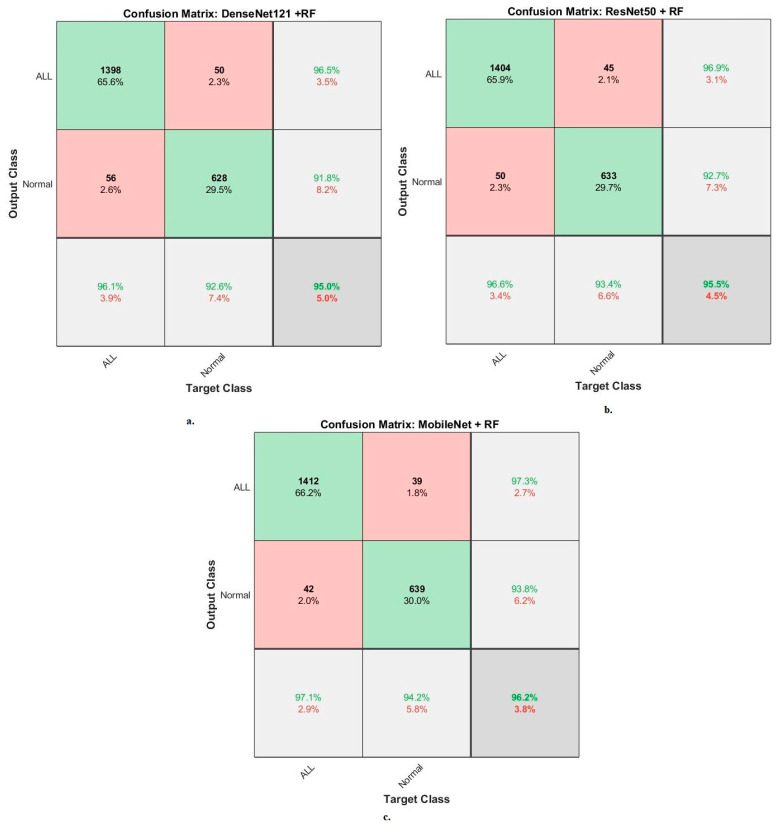
Results of the RF classifier based on the features of the CNN models of the C-NMC 2019 dataset for the diagnosis of acute leukemia confusion matrix of (**a**) DenseNet121-RF. (**b**) ResNet50-RF. (**c**) MobileNet-RF.

**Figure 11 diagnostics-13-01026-f011:**
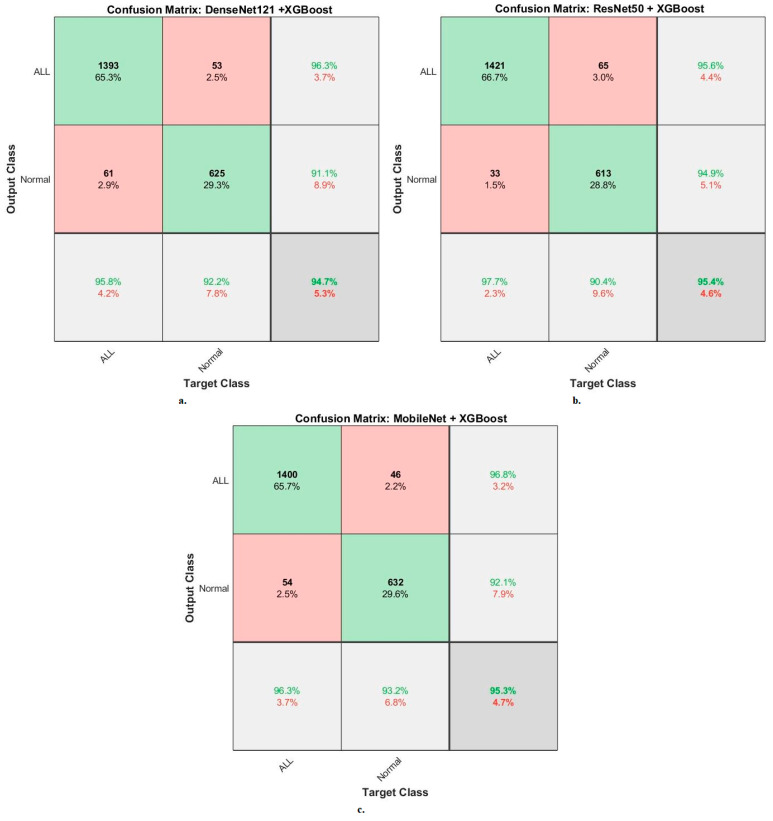
Results of the XGBoost classifier based on the features of the CNN models of the C-NMC 2019 dataset for the diagnosis of acute leukemia confusion matrix of (**a**) DenseNet121-XGBoost. (**b**) ResNet50-XGBoost. (**c**) MobileNet-XGBoost.

**Figure 12 diagnostics-13-01026-f012:**
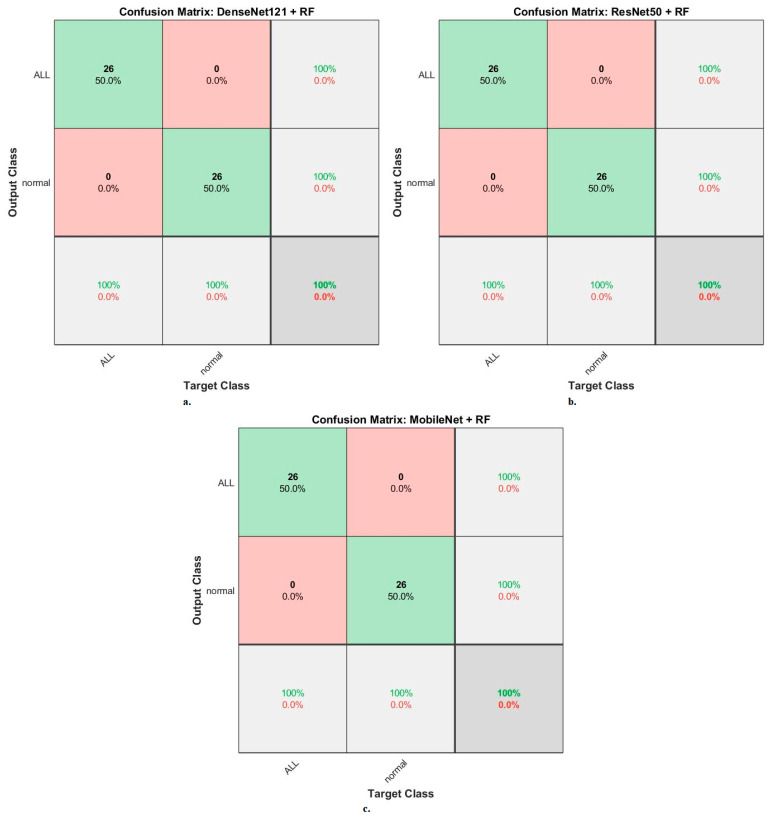
Results of the RF classifier based on the features of the CNN models of the ALL-IDB2 dataset for the diagnosis of acute leukemia confusion matrix of (**a**) DenseNet121-RF. (**b**) ResNet50-RF. (**c**) MobileNet-RF.

**Figure 13 diagnostics-13-01026-f013:**
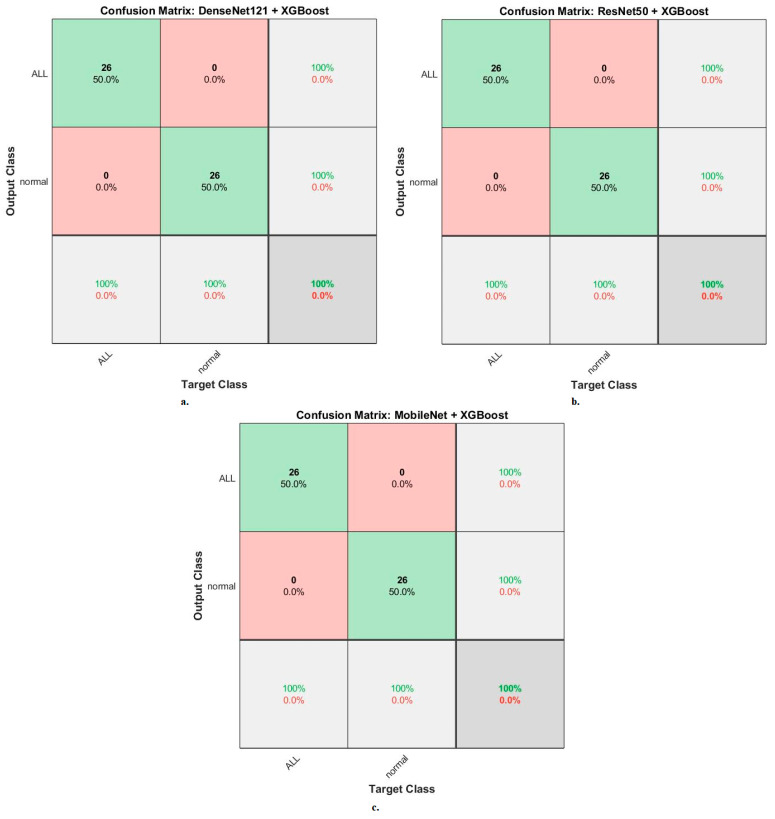
Results of the XGBoost classifier based on the features of the CNN models of the ALL-IDB2 dataset for the diagnosis of acute leukemia confusion matrix of (**a**) DenseNet121-XGBoost. (**b**) ResNet50-XGBoost. (**c**) MobileNet-XGBoost.

**Figure 14 diagnostics-13-01026-f014:**
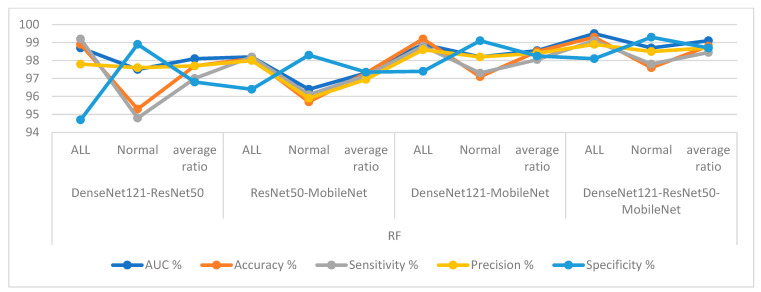
Display of results of the RF and XGBoost classifiers based on fusion features of the CNN models of the C-NMC 2019 dataset for the diagnosis of acute leukemia.

**Figure 15 diagnostics-13-01026-f015:**
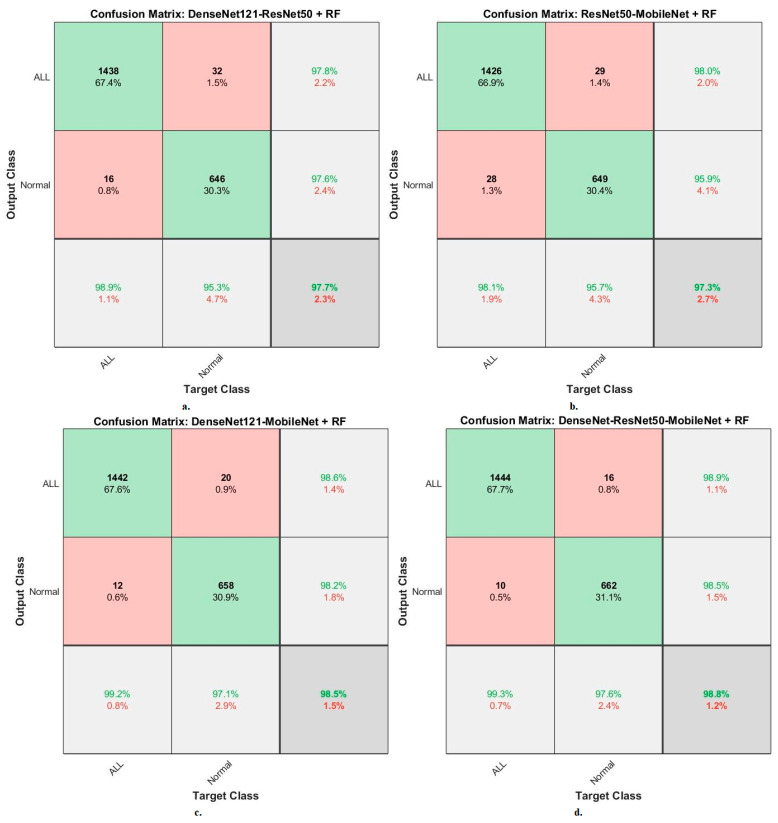
Results of the RF classifier based on fusion features of the CNN models of the C-NMC 2019 dataset for the diagnosis of acute leukemia, confusion matrix of (**a**) DenseNet121-ResNet50, (**b**) ResNet50-MobileNet, (**c**) DenseNet121-MobileNet, and (**d**) DenseNet121-ResNet50-MobileNet.

**Figure 16 diagnostics-13-01026-f016:**
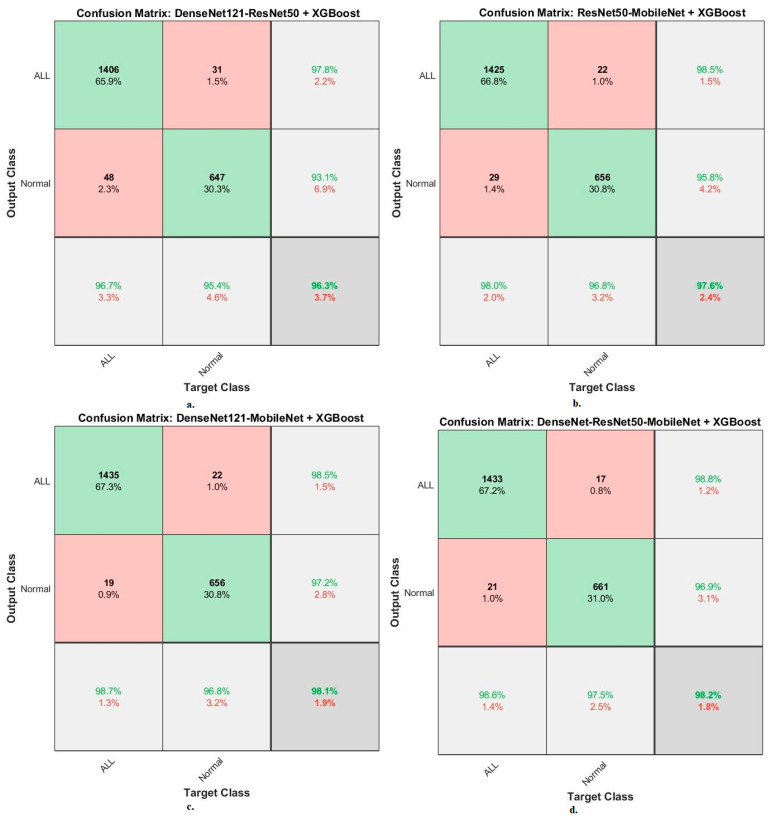
Results of the XGBoost classifier based on fusion features of the CNN models of the C-NMC 2019 dataset for the diagnosis of acute leukemia, confusion matrix of (**a**) DenseNet121-ResNet50, (**b**) ResNet50-MobileNet, (**c**) DenseNet121-MobileNet, and (**d**) DenseNet121-ResNet50-MobileNet.

**Figure 17 diagnostics-13-01026-f017:**
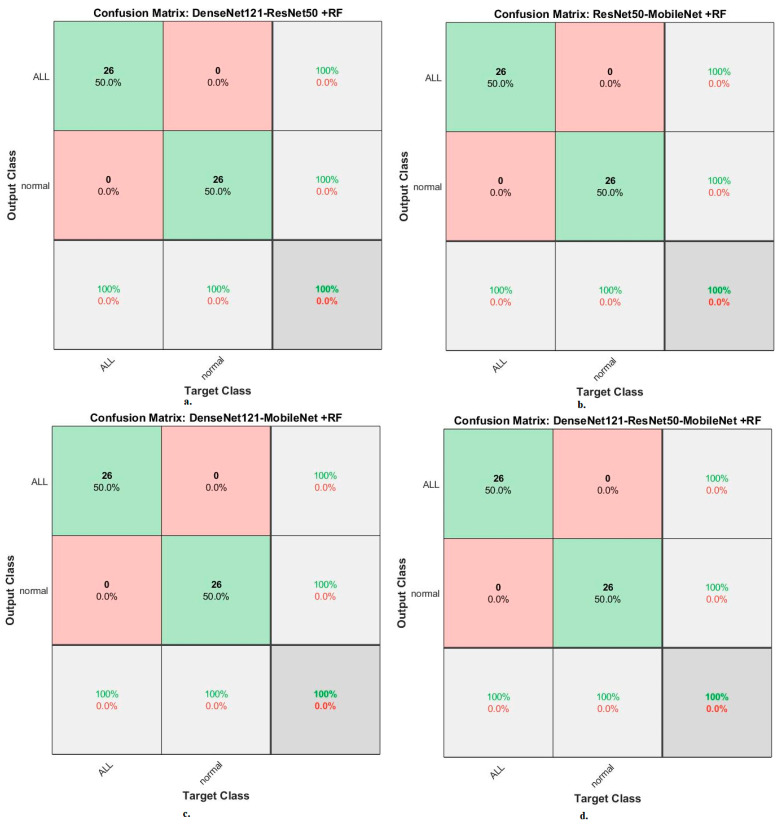
Results of the RF classifier based on fusion features of the CNN models of the ALL-IDB2 dataset for the diagnosis of acute leukemia, confusion matrix of (**a**) DenseNet121-ResNet50, (**b**) ResNet50-MobileNet, (**c**) DenseNet121-MobileNet, and (**d**) DenseNet121-ResNet50-MobileNet.

**Figure 18 diagnostics-13-01026-f018:**
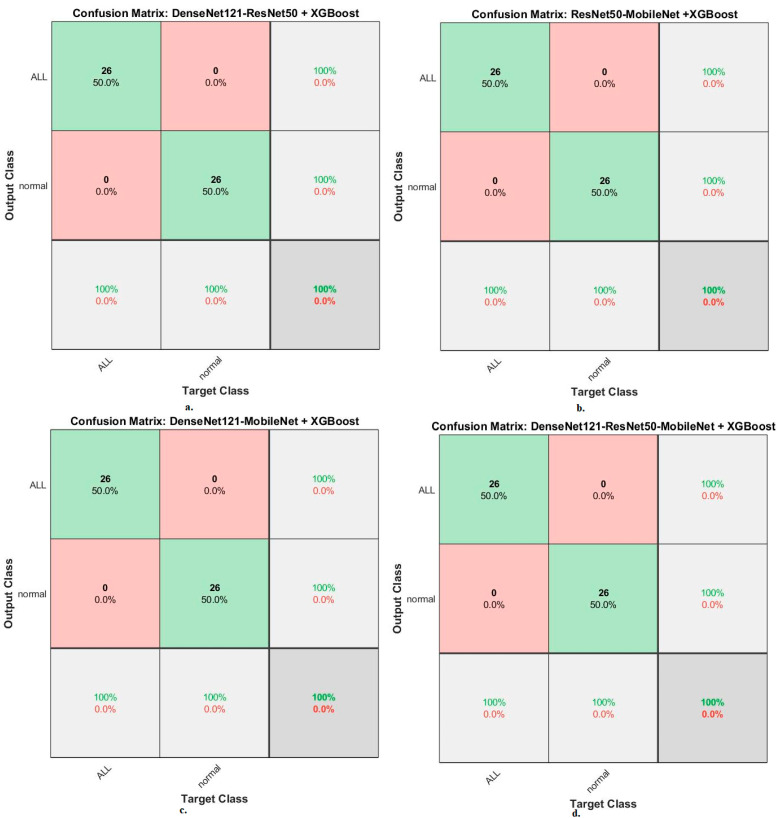
Results of the XGBoost classifier based on fusion features of the CNN models of the ALL-IDB2 dataset for the diagnosis of acute leukemia, confusion matrix of (**a**) DenseNet121-ResNet50, (**b**) ResNet50-MobileNet, (**c**) DenseNet121-MobileNet, and (**d**) DenseNet121-ResNet50-MobileNet.

**Figure 19 diagnostics-13-01026-f019:**
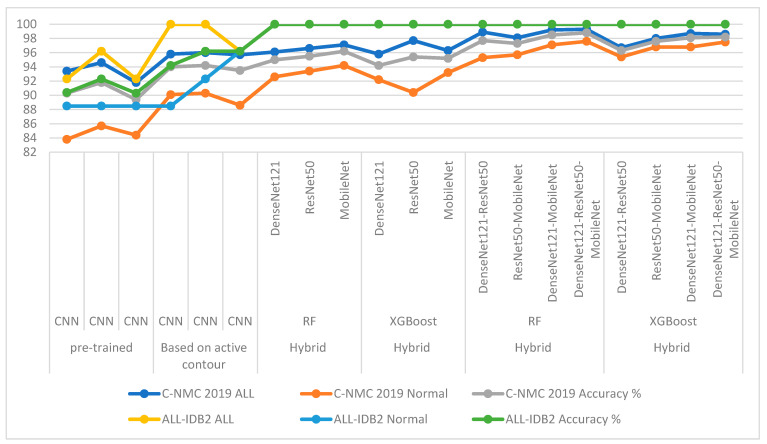
Displaying the results of all systems for the analysis of microscopic blood images for the C-NMC 2019 and ALL-IDB2 datasets acute leukemia at the level of each class.

**Table 1 diagnostics-13-01026-t001:** Splitting the C-NMC 2019 and ALL-IDB2 datasets.

Datasets		C-NMC 2019			ALL-IDB2		
Phase		80% (80:20)		Testing 20%	80% (80:20)		Testing 20%
	Phase	Training (80%)	Validation (20%)		Training (80%)	Validation (20%)	
Classes							
ALL		4654	1164	1454	83	21	26
Normal		2169	542	678	83	21	26

**Table 2 diagnostics-13-01026-t002:** Adjust hyperparameters options for DenseNet121, ResNet50, and MobileNet models.

Options	DenseNet	ResNet50	MobileNet
training Options	adam	adam	sgdm
Mini Batch Size	30	12	7
Max Epochs	20	4	25
Initial Learn Rate	0.0003	0.0001	0.0004
Validation Frequency	50	3	40
Execution Environment	4 GB GPU NVIDIA	4 GB GPU NVIDIA	4 GB GPU NVIDIA

**Table 3 diagnostics-13-01026-t003:** Data augmentation method of C-NMC 2019 and ALL-IDB2 datasets for acute leukemia.

Datasets	C-NMC 2019		ALL-IDB2	
Phase	Training Dataset		Training Dataset	
Classes	ALL	Normal	ALL	Normal
Bef-augmen	4654	2169	83	83
Aft-augmen	**9308**	**8676**	**1660**	**1660**

**Table 4 diagnostics-13-01026-t004:** Results of DenseNet121, ResNet50, and MobileNet models pre-trained on two datasets, C-NMC 2019 and ALL-IDB2, for acute leukemia diagnosis.

DataSets	Models	Classes	AUC %	Accuracy %	Sensitivity %	Precision %	Specificity %
C-NMC 2019	DenseNet121	ALL	95.2	93.4	93.2	92.5	83.8
		Normal	94.1	83.8	84.4	85.5	92.9
		**average ratio**	**94.65**	**90.3**	**88.8**	**89**	**88.35**
	ResNet50	ALL	94.7	94.6	95.2	92.8	86.1
		Normal	95.6	85.7	85.9	88.2	94.8
		**average ratio**	**95.15**	**90.8**	**90.55**	**90.5**	**90.53**
	MobileNet	ALL	92.9	91.8	92.2	92.6	84.2
		Normal	93.2	84.4	83.7	82.8	92.4
		**average ratio**	**93.05**	**89.4**	**87.95**	**87.7**	**88.3**
ALL-IDB2	DenseNet121	ALL	93.9	92.3	92.4	88.9	88.3
		Normal	93.7	88.5	87.9	92	92.1
		**average ratio**	**93.8**	**90.4**	**90.15**	**90.45**	**90.2**
	ResNet50	ALL	92.8	96.2	96.1	89.3	87.7
		Normal	91.9	88.5	88.4	95.8	95.9
		**average ratio**	**92.35**	**92.3**	**92.25**	**92.55**	**91.8**
	MobileNet	ALL	93.4	92.3	92.4	88.9	87.7
		Normal	92.7	88.5	88.1	92	92.2
		**average ratio**	**93.05**	**90.4**	**90.25**	**90.45**	**89.95**

**Table 5 diagnostics-13-01026-t005:** Results of CNN models based on segmentation WBC of C-NMC 2019 and ALL-IDB2 datasets for acute leukemia diagnosis.

DataSets	Models	Classes	AUC %	Accuracy %	Sensitivity %	Precision %	Specificity %
C-NMC 2019	DenseNet121	ALL	95.7	95.8	96.2	95.4	89.7
		Normal	96.2	90.1	90.4	90.9	95.8
		**average ratio**	**95.95**	**94**	**93.3**	**93.15**	**92.75**
	ResNet50	ALL	94.1	95.7	95.8	94.8	89.4
		Normal	95.7	88.6	90.4	90.6	96.3
		**average ratio**	**94.9**	**93.5**	**93.1**	**92.7**	**92.85**
	MobileNet	ALL	95.9	96	96.1	95.5	90.2
		Normal	97.1	90.3	90.4	91.3	95.9
		**average ratio**	**96.5**	**94.2**	**93.25**	**93.4**	**93.05**
ALL-IDB2	DenseNet121	ALL	97.1	100	99.5	89.7	88.1
		Normal	96.8	88.5	87.8	100	99.7
		**average ratio**	**96.95**	**84.2**	**93.65**	**94.85**	**93.9**
	ResNet50	ALL	96.8	100	99.8	92.9	91.9
		Normal	97.4	92.3	91.8	100	99.6
		**average ratio**	**97.1**	**96.2**	**95.8**	**96.45**	**95.75**
	MobileNet	ALL	96.8	96.2	96.3	95.9	96.4
		Normal	96.5	96.2	95.9	96.4	96.3
		**average ratio**	**96.65**	**96.2**	**96.1**	**96.15**	**96.35**

**Table 6 diagnostics-13-01026-t006:** Results of the RF and XGBoost classifiers based on the features of the CNN models of the C-NMC 2019 dataset for the diagnosis of acute leukemia.

Classifier	Models for Features	Classes	AUC %	Accuracy %	Sensitivity %	Precision %	Specificity %
RF	DenseNet121	ALL	98.4	96.1	96.2	96.5	92.9
		Normal	97.9	92.6	93.4	91.8	95.7
		**average ratio**	**98.15**	**95**	**94.8**	**94.15**	**94.3**
	ResNet50	ALL	97.6	96.6	97.3	96.9	92.9
		Normal	96.9	93.4	93.4	92.7	96.5
		**average ratio**	**97.25**	**95.5**	**95.35**	**94.8**	**94.7**
	MobileNet	ALL	98.2	97.1	97.2	97.3	94.2
		Normal	95.3	94.2	94.3	93.8	97.1
		**average ratio**	**96.75**	**96.2**	**95.75**	**95.55**	**95.65**
XGBoost	DenseNet121	ALL	95.9	95.8	96.4	96.3	91.9
		Normal	95.1	92.2	95.8	91.1	95.7
		**average ratio**	**95.5**	**94.7**	**96.1**	**93.7**	**93.8**
	ResNet50	ALL	98.9	97.7	97.9	95.6	90.4
		Normal	96.4	90.4	90.4	94.9	98.2
		**average ratio**	**97.65**	**95.4**	**94.15**	**95.25**	**94.3**
	MobileNet	ALL	97.1	96.3	95.5	96.8	93.4
		Normal	94.9	93.2	93.2	92.1	96.1
		**average ratio**	**96**	**95.3**	**94.35**	**94.45**	**94.75**

**Table 7 diagnostics-13-01026-t007:** Results of the RF and XGBoost classifiers based on fusion features of the CNN models of the C-NMC 2019 dataset for the diagnosis of acute leukemia.

Classifier	Fusion Features	Classes	AUC %	Accuracy %	Sensitivity %	Precision %	Specificity %
RF	DenseNet121-ResNet50	ALL	98.7	98.9	99.2	97.8	94.7
		Normal	97.5	95.3	94.8	97.7	98.9
		**average ratio**	**98.1**	**97.7**	**97**	**97.75**	**96.8**
	ResNet50-MobileNet	ALL	98.4	98.1	98.2	98	96.4
		Normal	96.4	95.7	96.1	95.9	98.3
		**average ratio**	**97.4**	**97.3**	**97.15**	**96.95**	**97.35**
	DenseNet121-MobileNet	ALL	98.9	99.2	98.8	98.6	97.4
		Normal	98.2	97.1	97.3	98.2	99.1
		**average ratio**	**98.55**	**98.5**	**98.05**	**98.4**	**98.25**
	DenseNet121-ResNet50-MobileNet	ALL	99.5	99.3	99.1	98.9	98.4
		Normal	98.7	97.6	97.8	98.5	99.3
		**average ratio**	**99.1**	**98.8**	**98.45**	**98.7**	**98.85**
XGBoost	DenseNet121-ResNet50	ALL	97.9	96.7	97.4	97.8	94.5
		Normal	96.2	95.4	94.8	93.1	97.1
		**average ratio**	**97.05**	**96.3**	**96.1**	**95.45**	**95.8**
	ResNet50-MobileNet	ALL	98.1	98	98.2	98.5	96.8
		Normal	97.6	96.8	97.4	95.8	97.9
		**average ratio**	**97.85**	**97.6**	**97.8**	**97.15**	**97.35**
	DenseNet121-MobileNet	ALL	99.2	98.7	98.7	98.5	96.8
		Normal	98.5	96.8	96.9	97.2	99.1
		**average ratio**	**98.85**	**98.1**	**97.8**	**97.85**	**97.95**
	DenseNet121-ResNet50-MobileNet	ALL	99	98.6	99.2	98.8	97.4
		Normal	98.1	97.5	96.8	96.9	98.9
		**average ratio**	**98.55**	**98.2**	**98**	**97.85**	**98.15**

**Table 8 diagnostics-13-01026-t008:** Results of all proposed systems for diagnosing images of the NMC 2019 and ALL-IDB2 datasets based on hybrid technologies and fused features.

Datasets			C-NMC 2019			ALL-IDB2		
Techniques		Features	ALL	Normal	Accuracy %	ALL	Normal	Accuracy %
pre-trained	DenseNet121		93.4	83.8	90.3	92.3	88.5	90.4
	ResNet50		94.6	85.7	91.8	96.2	88.5	92.3
	MobileNet		91.8	84.4	89.4	92.3	88.5	90.3
Based on active contour	DenseNet121		95.8	90.1	94	100	88.5	94.2
	ResNet50		96	90.3	94.2	100	92.3	96.2
	MobileNet		95.7	88.6	93.5	96.2	96.2	96.2
Hybrid	RF	DenseNet121	96.1	92.6	95	100	100	100
		ResNet50	96.6	93.4	95.5	100	100	100
		MobileNet	97.1	94.2	96.2	100	100	100
Hybrid	XGBoost	DenseNet121	95.8	92.2	94.2	100	100	100
		ResNet50	97.7	90.4	95.4	100	100	100
		MobileNet	96.3	93.2	95.2	100	100	100
Hybrid	RF	DenseNet121-ResNet50	98.9	95.3	97.7	100	100	100
		ResNet50-MobileNet	98.1	95.7	97.3	100	100	100
		DenseNet121-MobileNet	99.2	97.1	98.5	100	100	100
		DenseNet121-ResNet50-MobileNet	99.3	97.6	98.8	100	100	100
Hybrid	XGBoost	DenseNet121-ResNet50	96.7	95.4	96.3	100	100	100
		ResNet50-MobileNet	98	96.8	97.6	100	100	100
		DenseNet121-MobileNet	98.7	96.8	98.1	100	100	100
		DenseNet121-ResNet50-MobileNet	98.6	97.5	98.2	100	100	100

## Data Availability

In this study, data supporting the performance of the proposed systems for early diagnosis of acute leukemia were collected from C-NMC 2019 and ALL-IDB2 datasets that are publicly available at the links: https://wiki.cancerimagingarchive.net/pages/viewpage.action?pageId=52758223; https://www.kaggle.com/nikhilsharma00/leukemia-dataset (17 November 2022).

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
