# Peer review of "Hybrid Techniques for the Diagnosis of Acute Lymphoblastic Leukemia Based on Fusion of CNN Features"

_diagnostics, 2023, doi:10.3390/diagnostics13061026_

Round 1

Reviewer 1 Report

In this study, Hybrid Techniques for the Diagnosis of Acute Lymphoblastic Leukemia based on deep feature extraction is proposed. The study content is interesting. However, there are some major shortcomings. These shortcomings need to be corrected. These are given below:

1- The novel aspects of the study should be highlighted.

2- The literature review needs to be improved. You can examine some of the example studies given below.

- Automated detection of pain levels using deep feature extraction from shutter blinds-based dynamic-sized horizontal patches with facial images
- A discriminatively deep fusion approach with improved conditional GAN (im-cGAN) for facial expression recognition
- PatchResNet: Multiple Patch Division-Based Deep Feature Fusion Framework for Brain Tumor Classification Using MRI Images

7- Limitations of the proposed method should be given.

8- Conclusion section should be improved. Perspectives for future studies should be presented.

9- There are some spelling mistakes in the study. It would be appropriate to review and revise the study again.

Reviewer 2 Report

Recommendation: Major Revision

After the authors clearly respond to the all comments, the reviewer will reconsider acceptance.

The authors proposed effective strategies for early diagnosis of acute leukemia by analyzing the 727 images of C-NMC 2019 and ALL-IDB2 datasets leukemia.

1. Describe the data set used in the paper clearly in the form of a table with the number of data and the number of features.

2. The biggest advantage of using deep learning is that it automatically reduce dimensions while extracting important features. Please describe more in detail the reason why the authors applied PCA, which is a traditional method after applying deep learning-based CNN model.

3. Literature review is incomplete. The authors should provide reviews of recently machine learning-based models for biological problems, such as the paper with PMIDs: 35849666, 35743670 and https://doi.org/10.1016/j.knosys.2023.110295 .

4. Detailed information on hyperparameters and how they were tuned should be described so that readers can re-implement the model easily just by reading the paper.

5. The authors should revise English writing carefully and eliminate small errors in the paper to make the paper easier to understand.

Round 2

Reviewer 2 Report

Still, machine learning-based models are not discussed yet. The authors should discuss the aforementioned articles:

https://doi.org/10.3389/fncom.2022.1083649
https://doi.org/10.1109/TCBB.2022.3191972
https://doi.org/10.3390/jpm12060885
https://doi.org/10.1158/2643-3230.BCD-21-0095
https://doi.org/10.3390/cancers12071883

Round 3

Reviewer 2 Report

Authors handled comments well. Accept this form without any changes.